# The Human TOR Signaling Regulator Is the Key Indicator of Liver Cancer Patients’ Overall Survival: TIPRL/LC3/CD133/CD44 as Potential Biomarkers for Early Liver Cancers

**DOI:** 10.3390/cancers13122925

**Published:** 2021-06-11

**Authors:** Soo Young Jun, Hyang Ran Yoon, Ji-Yong Yoon, Su-Jin Jeon, Jeong-Ju Lee, Debasish Halder, Jin-Man Kim, Nam-Soon Kim

**Affiliations:** 1Medical Genomics Research Center, Korea Research Institute of Bioscience & Biotechnology, Daejeon 34141, Korea; jyoung@kribb.re.kr (S.Y.J.); sererine@kribb.re.kr (J.-Y.Y.); sjjeon@kribb.re.kr (S.-J.J.); snailee@kribb.re.kr (J.-J.L.); debasish@organoidrx.com (D.H.); 2Rare-Disease Research Center, Korea Research Institute of Bioscience & Biotechnology, Daejeon 34141, Korea; 3Department of Functional Genomics, University of Science & Technology, Daejeon 34113, Korea; 4Immunotherapy Convergence Research Center, Korea Research Institute of Bioscience & Biotechnology, Daejeon 34141, Korea; yhr1205@kribb.re.kr; 5Department of Pathology, Chungnam National University College of Medicine, Chungnam National University Hospital, 266 Munhwa-ro, Jung-gu, Daejeon 35015, Korea; jinmank@cnu.ac.kr

**Keywords:** liver cancer, human TOR signaling regulator (TIPRL), microtubule-associated light chain 3 (LC3), prominin-1 (CD133), cluster of differentiation 44 (CD44), receiver-operating characteristic (ROC) curve, Kaplan–Meier analysis, uni-/multi-Cox analyses, hepatocellular carcinomas (HCCs), intrahepatic carcinomas (iCCA)

## Abstract

**Simple Summary:**

We recently reported that the human TOR signaling regulator (hereafter TIPRL) contributes to the drug-resistance of hepatocellular carcinomas (HCCs) and the involvement of TIPRL/LC3/CD133 in liver cancer aggressiveness. This study aims to determine prognostic and diagnostic efficacies of TIPRL/LC3/CD133/CD44 for early liver cancer. We observed the significant upregulation of TIPRL and LC3 in HCCs and adult hepatocyte-derived liver disease while observing downregulation in intrahepatic carcinomas (iCCA). The TIPRL level has been shown to be the clearest indicator of liver cancer patients’ survivability as a sole covariate. This indication supports that TIPRL contributed to liver cancer cell proliferation and survival via stemness and self-renewal induction. TIPRL/LC3/CD133 have exhibited crucial efficiency in diagnostic patients with grade 1 iCCA, and TIPRL/LC3/CD133/CD44 showed prognosticating grade-1 HCCs and iCCA, either as an alone or in conjunction. Overall, this study reports that TIPRL/LC3/CD133/CD44 could, either individually or in conjunction, serve as potential biomarkers for early liver cancer.

**Abstract:**

Recently, we reported the involvement of TIPRL/LC3/CD133 in liver cancer aggressiveness. This study assessed the human TOR signaling regulator (TIPRL)/microtubule-associated light chain 3 (LC3)/prominin-1 (CD133)/cluster of differentiation 44 (CD44) as potential diagnostic and prognostic biomarkers for early liver cancer. For the assessment, we stained tissues of human liver disease/cancer with antibodies against TIPRL/LC3/CD133/CD44/CD46, followed by confocal observation. The roles of TIPRL/LC3/CD133/CD44/CD46 in liver normal and cancer cell lines were determined by in vitro studies. We analyzed the prognostic and diagnostic potentials of TIPRL/LC3/CD133/CD44/CD46 using the receiver-operating characteristic curve, a Kaplan–Meier and uni-/multi-Cox analyses. TIPRL and LC3 were upregulated in tissues of HCCs and adult hepatocytes-derived liver diseases while downregulated in iCCA. Intriguingly, TIPRL levels were found to be critically associated with liver cancer patients’ survivability, and TIPRL is the key player in liver cancer cell proliferation and viability via stemness and self-renewal induction. Furthermore, we demonstrate that TIPRL/LC3/CD133 have shown prominent efficiency for diagnosing patients with grade 1 iCCA. TIPRL/LC3/CD133/CD44 have also provided excellent potential for prognosticating patients with grade 1 iCCA and grade 1 HCCs, together with demonstrating that TIPRL/LC3/CD133/CD44 are, either individually or in conjunction, potential biomarkers for early liver cancer.

## 1. Introduction

Liver cancer is responsible for the second-most cancer-related deaths worldwide [1]. Despite rapid advances in diagnosis and therapy, liver cancer death rates have increased by 2% per year since 2000. The World Health Organization (WHO) predicts more than one million patient deaths from liver cancer in 2030 [2]. Public health agencies, such as WHO, see the late-stage presentation because most patients do not feel any signs and symptoms until liver cancers develop into the late stages and inaccessible diagnosis and treatment as the cause of this increase in deaths caused by liver cancer [1]. Accordingly, there is an urgent demand to develop diagnostic resources, including a validated tumor-marker or -panels to improve the diagnosis of early liver cancers before they transform into an overt malignant phenotype.

Previously, our team demonstrated that the human TOR signaling regulator protein (hereafter TIPRL) contributes to tumor necrosis factor-related apoptosis-inducing ligand (TRAIL) resistance of hepatocellular carcinomas (HCCs) via the negative regulation of the MKK7/JNK pathway [3]. TIPRL is the mammalian ortholog of yeast TIP41, a binding partner for the type 2A-associated protein of 42 kilodaltons (Tap42) [4]. Unlike yeast TIP41 binding to α4, TIPRL interacts directly with the protein phosphatase (PP) type 2Ac, or PP type 4, or PP type 6, thereby inhibiting PP’s activity suppressing ataxia telangiectasia mutated (ATM), and ATM and Rad3-related (ATR) dependent phosphorylation events [4,5]. Consequently, the interaction between TIPRL and PP controls the mammalian target of rapamycin (mTOR) activity [6]. Regarding this pathway [6], we recently reported that TIPRL enhances cancer cell survival in metabolic and cellular stress via the induction of autophagic clearance [7]. TIPRL accelerates liver cancer aggressiveness via the upregulation of the microtubule-associated protein light chain 3 (LC3), an autophagy marker, and CD133 (Prominin-1) expression [8].

The differentiation 44 (CD44) cluster, a hyaluronic acid receptor, is expressed in many different types of cancer-initiating cells, cancer stem cells (CSCs), and rapidly proliferating cells. Its expression represents increased proliferation, self-renewal, and metastasis of cancer cells [9]. For instance, CD44high/CD133high cells exhibited increased tumorigenic capabilities [10]. In addition, CD44 has been used for CSC identification in various malignancies, including HCCs, together with CD90, CD133, CD24, and EpCAM [11]. CD133 is found on human embryonic stem cells (hESCs) and rarely on normal tissue cells. CD133 is also used as a CSC marker to characterize cells with high tumorigenicity and the ability to form spheroids [12]. Chen et al. reported that the active re-location of CD133 from the plasma membrane into the cytoplasm induces autophagy, demonstrating a strong association with CD133 and LC3 [13]. This suggests that CD133 has a critical role in the induction of autophagy in HCC cells.

During the autophagy process, protein and damaged or aged organelles were sequestered and degraded in autophagosomes, and were then recycled for energy production and new protein and membrane construction [14]. Autophagy has reported its dual role in cancer, such as a tumor suppressor via the deletion of damaged mitochondria (mitophagy) [15] or as a tumor promoter via the reduction of p53 expression [9]. The production of hepatic CSCs through autophagy induction exacerbates liver malignancy [16], and CSCs are considered the cause of tumor initiation and relapses. Therefore, the discovery of biomarkers identifying cells or CSCs carrying a risk for disease progression and subsequent post-therapeutic relapses is indispensably required. In this study, we examined the clinical implications of the variables, TIPRL, LC3, CD133, and CD44, all of which have been reported on in terms of their roles in liver cancers and autophagy, including hepatocellular carcinomas (HCCs) and intrahepatic cholangiocarcinomas (iCCA). We also attempted to determine the relationships of these variables in liver cancer aggressiveness. Here we report the crucial role of TIPRL in liver cancer patients’ survivability and liver cancer cells’ survival. Furthermore, our study provides evidence for the significant prognostic and diagnostic efficiencies of TIPRL/LC3/CD133/CD44, either individually or in conjunction, to detect early liver cancer.

## 2. Results

### 2.1. Inverse Expression of TIPRL and LC3 in HCCs and iCCA

To further extend our previous reports that TIPRL contributes to the TRAIL resistance of HCCs [3] and to the aggressiveness of HCCs via positive regulation of LC3 and CD133 expression [8], we examined levels of all five variables, TIPRL, LC3, CD133, CD44, and CD46, previously reported to contribute to chemo- and radio-resistance in liver cancers. For this, we stained human liver disease tissues, including HCCs and iCCA, with the indicated antibodies and then determined the level of variables (Appendix A) after performing global normalization using raw data obtained by confocal observation and the ZEN program (Figure 1A,C). We observed that the HCCs, belonging to cancer grade 2, exhibit mild pleomorphism and evident nucleoli (Figure 1B). The iCCA, morphologically cholangiolar-type iCCA, show glandular (upper) and cribriform (lower) forms (Figure 1D).

Furthermore, consistent with our previous report, we confirmed the significant upregulation of TIPRL and LC3 in HCCs, as demonstrated in our study and public DB (www.oncomine.org accessed on 5 January 2021), compared with the adjacent normal tissues (Figure 1A,B and Appendix A). Contrarily, we observed the substantial downregulation of TIPRL and LC3 in iCCA, CD133 and CD46 in both HCCs and iCCA, and CD44 in HCCs (Figure 1 and Appendix A). However, we failed to observe the upregulation of CD133 in HCCs as the sets used in this study contained significant percentages of G1 tissues (G1, 149/total, 1439 = 10.4%), while the set used in the previous report [8] did not have any G1. Besides, the level of CD133 in G1 was significantly lower than the level in normal liver tissues, and CD133 exhibited a grade-dependent increase in its expression pattern, as demonstrated in our cohort study and Public DB (www.oncomine.org accessed on 29 January 2021) (Appendix A, respectively).

In addition, we determined that TIPRL increases LC3, CD133, and CD44, except CD46, in a grade-dependent manner in HCCs and iCCA group (Appendix A, respectively). This grade-dependent increase was further supported by the observation that, compared to the low TIPRL expression, LC3 and CD133 expression were relatively upregulated in the high TIPRL expression group (Appendix A), together suggesting that TIPRL works as an upstream modulator of LC3, CD133, and CD44, which is consistent with our previous report [8].

Given our previous report on the involvement of TIPRL, LC3, and CD133 in human liver cancers [8], we further examined the relationships between the levels in all five variables in liver disease tissues. We observed a significant upregulation of TIPRL, LC3, and CD44 in tissues of hepatocyte-derived liver diseases such as chronic hepatitis, hepatic steatosis, liver cell degeneration, liver tissue degeneration, and inflammation of the porta area, compared with normal tissues. Contrarily, in cirrhosis, the end stage of liver fibrosis, TIPRL, LC3, and CD44 were downregulated. CD133 showed its upregulation in tissues of chronic hepatitis, hepatic steatosis, liver cell degeneration, and inflammation of the porta area (Appendix A). When examined in cholangiocarcinoma-derived diseases, such as liver carcinoid, the mixture of extra-and intra-cholangiocarcinomas, the four variables exhibited significant upregulation compared with the normal tissues, unlike iCCA. On the other hand, in adenosquamous carcinomas derived from the extrahepatic bile duct, TIPRL, LC3, and CD46 did not show a meaningful change, while CD133 exhibited significant upregulation compared to normal tissues (Appendix A). Overall, we determined the inverse expression of TIPRL and LC3, the positive regulators of autophagy, in HCCs and iCCA are derived from hepatocytes and bile ducts, respectively. In keeping with the upregulations in HCCs, TIPRL and LC3 showed significant increases in the adult hepatocytes-derived liver diseases, except for cirrhosis and adenosquamous carcinomas, compared to normal tissues. On the other hand, except for cirrhosis and liver tissue degeneration, we determined the upregulation of CD133, CD44, and CD46 in liver disease tissues. Therefore, our data suggests that, given the reports that the adult hepatocytes could be differentiated into HCCs and iCCA while mature cholangiocytes only are into iCCA, the level of all five variables, TIPRL, LC3, CD133, CD44, and CD46, is associated with the disease status of the liver.

### 2.2. The Significant Association between TIPRL and Liver Disease Patients’ Survival

Next, using the public database (www.kmplot.com accessed on 12 January 2021) and our training and validation sets (Appendix A), we examined the relationships between liver disease patients’ overall survival (OS) and the levels of all five variables.

Intriguingly, we observed that our training (Figure 2A) and validation (Appendix A) sets verified the significance of TIPRL in OS of liver disease patients’ in the public DB (Figure 2B). Furthermore, our data showed a significant hazard ratio (HR) of CD133 in both sets and LC3 and CD46 in the validation set in OS of liver disease patients (Appendix A), unlike the public DB containing only liver cancer patients (Figure 2B). Considering these differences, we further studied the association between disease-specific survival (DSS) of liver disease patients and the levels of four variables without CD46 (Appendix A). DSS determines the patient’s survival percentage in a defined time, which usually starts at diagnosis and ends at the time of death [17].

With the public DB, we observed that TIPRL exhibited the most significant HR ratio within acceptable ranges with a 95% confidence ratio (CI) and *p*-value in the stage 4 cohort (Appendix A). Additionally, in sorafenib-treated liver cancer patients, TIPRL, LC3, and CD133, showed a substantial HR and 95% CI, except for CD44 (Appendix A). We carefully think that these significant increases might be ascribed from the upregulated expression of TIPRL, LC3, and CD133 by chemotherapy treatment, such as TRAIL [18]. In keeping with the significance of TIPRL observed in the public DB, our data showed that TIPRL also had a significant HR on liver cancer patients’ DSS (Appendix A). Additionally, we demonstrated the meaningful relationship between the CD44 and DSS of liver cancer patients, unlike the public DB. Overall, our data suggests that TIPRL has the most significant effect on liver cancer patients’ survival.

### 2.3. The Significant Predictive Ability of TIPRL on Liver Disease Patients

We confirmed significant differences between the training (*n* = 790) and the validation (*n* = 649) set (Appendix A). The training set contained a substantial percentage of males and HCCs, while the validation had three times more iCCA than HCCs. Moreover, the training set had a considerable proportion of stage II, T2, and N0, while the validation had stage IV, T3, and N1 tissues. There was no significant difference in the OS of liver disease patients.

Next, to correct for confounding variables and then determine the independent effects of risk factors in liver disease patients, we merged the continuous variables and the clinical parameters. Then, a univariate Cox proportional hazard regression analysis was performed. We determined a reasonable *p*-value in the categorical factors, subtypes, and the continuous variables, TIPRL and CD133, on the OS of liver disease patients in the validation set (Table 1), even though the factors and variables in the training set have comparable values in HR and 95% CI (Table 1). We then carried out a multivariate analysis with the continuous variables and the significant categorical factors determined by the univariate analysis. The importance of TIPRL and CD133 on the OS of liver disease patients in the training set (Table 2) substantially increased in the validation, as shown in Table 2. We also determined the significance of CD44 on the OS of liver disease patients in the training set (Table 2), although not validated in the validation set. Regarding these significances, the variables, LC3, CD133, CD44, and TNM in the training set (Appendix A), and TIPRL, CD133, CD46, and Grade, in the validation set (Appendix A) showed significant independence. However, the independence of TIPRL failed on the OS of liver disease patients in the training set, as demonstrated by their *p*-values being higher than α-0.05 in the proportionality test.

To further evaluate the independence of TIPRL for the overall survival of the patients, we subdivided the training and the validation set into the training, with only HCCs and the validation, with only HCCs, and with only iCCA set. The univariate analysis revealed that all five sets show the significant association of TIPRL with OS, indicated by the HR within a similar range (Appendix A). Noticeably, we determined the similarities in sample distribution in the training set with only HCCs and the validation with only iCCA patients; moreover, the validation set confirms the association of the variables with the OS of the patients in the training set, although the LC3 failed to gain reasonable *p*-values. Additionally, we determined the significant independence of the two sets in the validation cohorts using the proportionality test, including the validation (TIPRL, *p* = 0.5) and with only HCCs patients, unlike the validation contains HCCs and iCCA patients (TIPRL, *p* = 0.071 and 0.041, respectively).

Overall, we believe that the differences in the HR of the variables between the training and the validation (Table 1), and why the TIPRL failed to gain independence in the training set (Appendix A) were ascribed to the different sample distribution, as exemplified by the categorical variable stage and composition in the training set, compared with the validation set.

### 2.4. TIPRL as a Key Player for Liver Cancer Survival

Given the variables’ relationship with the OS of liver cancer patients, we further examined the roles of all five variables, TIPRL, LC3 (ATG7), CD133, CD44, and CD46, in normal liver (Chang), HCC (huh7), and iCCA (SNU1097) cell lines. Interestingly, we observed that only two different siRNAs-TIPRL transfections significantly reduced HCC and iCCA cell proliferation, cultured in an attached, and cell viability, in an Anoikis condition than the other siRNAs-transfection did (Figure 3A,B; Appendix A). In line with these reductions, TIPRL knockdown decreased the expression of LC3, CD133, and CD44 (Appendix A), confirming our previous report [8].

Furthermore, we confirmed that, in contrast to the siRNAs-treated Chang cells (Figure 3C and Appendix A), the significant downregulation in mRNA expressions of *OCT4, Nanog, SOX2,* and *LIN28,* stemness-related genes, in siTIPRL-transfected huh7 and SNU1097 cells, consistent with our previous report (Figure 3D; Appendix A). Noticeably, even though CD133 knockdown substantially decreased the expression of stemness-related genes be comparable to TIPRL knockdown in huh7 and SNU1097 cells, only TIPRL depletion significantly reduced cell viability and the self-renew ability of HCCs and iCCA (Figure 3B,E,F; Appendix A), confirming our previous report that TIPRL works as an upstream for LC3 and CD133 [8], and thereby contributes to liver cancer cell proliferation, viability, and stemness, which are key events for liver cancer incidence/progression.

### 2.5. TIPRL, LC3, CD133, and CD44 as Liver Cancer Biomarkers for Early Diagnosis and Prognosis

Next, we analyzed the diagnostic and prognostic potentials of all five variables in human liver disease (Appendix A). The receiver operating characteristic (ROC) analysis showed an approximately 50% potential of area under the curve (AUC) value in each model and a 62.8% AUC value of CD44. These AUC values are unsatisfactory areas for diagnosis [19]. When analyzed using the Youden index, we failed to observe an increase in sensitivity and specificity of the following combined biomarkers: TIPRL/LC3/CD133, TIPRL/LC3/CD133/CD44, TIPRL/LC3/CD133/CD44/CD46 (Appendix A). Contrarily, when we investigated the ability of all five variables on the prognosis of liver disease patients, our data demonstrated a substantial ratio of AUC with acceptable ranges of *p*-value in each model. Moreover, the combined models exhibited a significant increase in sensitivity with a reasonable 95% CI. However, we failed to observe any increased effect on sensitivity and specificity when calculated based on the Youden index (Appendix A). Together, our data suggests a crucial prognostic potential of all five variables on liver disease patients.

In agreement with our previous report, we demonstrated a significant association between the levels in all five variables in human liver cancer tissues (Appendix A). Considering the significant prognosticating (Appendix A) and association of levels in the candidates in liver cancer tissues (Appendix A), we attempted to analyze the diagnostic and prognostic potentials of all five variables in liver cancer patients in the training and validation sets.

Intriguingly, we noticed the training set containing only HCCs exhibited 10–20% higher AUC values than the validation containing HCCs and iCCA in a diagnostic analysis (Appendix A). Moreover, the combined biomarkers, TIPRL/LC3/CD133/CD44, in the training set massively reduced the required detection amount and enhanced sensitivity with an acceptable range of 95% CI and *p*-values (Appendix A). When we examined the prognostic potentials, unlike the diagnostic ability, both sets exhibited a similar AUC ratio, although the training set showed a 10% higher sensitivity than the validation set. We failed to find any effect of combining biomarkers in evaluating AUC and the sensitivity as the observed effects in the diagnostic evaluation (Appendix A). Overall, our data provides evidence that the variables, TIPRL, LC3, CD133, and CD44, have a prominent effect on a prognostic role in liver disease/cancer patients (Appendix A). Furthermore, these biomarkers are more efficient at detecting HCCs than a mixture of HCCs and iCCA (Appendix A).

Early detection of cancers significantly improves the survival of patients [20]. Furthermore, considering the reports that patients with early-stage liver cancer have more than a 33% five-year survival rate, while patients with advanced stages have less than 2% [21], we analyzed the early diagnostic and prognostic potentials of all five variables in HCCs and iCCA tissues. Interestingly, when investigating the diagnostic ability in grade 1 iCCA, we observed a 15–25% increase in AUC ratio with an acceptable range of 95% CI and *p*-value (Figure 4A,B) in every model, except for CD46, compared to all grades of iCCA (Appendix A). We also demonstrated a 5–19% increase in the AUC ratio with a reasonable range of 95% CI and *p*-value in prognosis analysis of grade 1 iCC tissues (Figure 4C,D) compared to all grades of iCCA (Appendix A). Furthermore, our data shows the substantial efficiency of TIPRL in diagnosis and prognosis abilities of grade 1 level iCCA (75.2% and 91.1%, respectively; Figure 4A,C). In addition, LC3 and CD133 in the prognostic of grade 1 level iCCA tissues showed a significant AUC ratio (87.5% and 78.6%, respectively; Figure 4C,D). On the contrary, compared to all grade levels of HCCs (Appendix A), there was no significant increase in diagnostic ability in grade 1 HCC (Appendix A), consistent with our previous report. However, compared to the whole grade of HCCs (Appendix A), we still observed a substantial range of AUC ratios in the prognosis of grade 1 HCC tissues (Appendix A). Additionally, we determined, even though prognostic and diagnostic efficiencies were reduced according to cancer grade, the statistically acceptable prognostic efficiency with sensitivity in cancer grade 2/3 (Appendix A). Overall, our data provides evidence that the variables without CD46 are suitable for diagnostic patients with grade 1 iCCA and superior for prognosticating with the grade 1 iCCA and the grade 1 HCCs, and can also be appropriate for cancer grade 2/3 iCCA/HCCs patients.

## 3. Discussion

The average five-year survival rate for liver cancer patients is less than 6% worldwide. This low rate is due to the majority of patients being diagnosed at advanced stages: 43% diagnosed at localized, 27% at regional, 18% at distant, and 12% at an unstaged status (five-year survival rates are 31.1%, 10.1%, 2.8%, and 6.4%, respectively; SEER website) [22]. Furthermore, a recent steady increase in the incidence of liver cancer—over 2% per year—has pushed us to discover potential targets for early detection of primary liver cancers with acceptable sensitivity and specificity that can be translated into the clinic. Here, we demonstrate that TIPRL and LC3 are upregulated and downregulated in hepatocellular carcinomas (HCCs) and intrahepatic cholangiocarcinomas (iCCA), respectively. Furthermore, the level of TIPRL is significantly related to liver disease and cancer patients’ overall survival. TIPRL has a critical role in liver cancer cell survival via stemness and self-renewal induction. Considering our previous reports that TIPRL contributes to liver cancer aggressiveness via the modulation of LC3 and CD133 [8], the crucial roles of TIPRL in liver disease/cancer further support the demonstration that the variables, TIPRL, LC3, CD133, and CD44, are suitable covariates for diagnostic patients with grade 1 iCCA and prognosticating with the grade 1 iCCA and the grade 1 HCCs.

While adult cholangiocytes provide the only iCCA due to their lack of plasticity and transforming capacity, adult hepatocytes are the source of both HCCs and iCCA as the adult hepatocytes can directly degenerate into HCCs and indirectly dedifferentiate into mature hepatocytes. Then, the hepatocytes transdifferentiate into biliary-like cells and later transform into iCCA [23]. In this study, we determined up- and down-regulation of TIPRL and LC3 in HCCs and iCCA, respectively. However, regardless of the TIPRL level in HCCs and iCCA cells, TIPRL knockdown decreased the expression of LC3, CD133, and CD44. Given the reports that iCCA are mainly derived from the Notch/Akt-mediated conversion of hepatocytes [24,25], we carefully reason that this pathway contributes to the more severe malignant potential of iCCA than HCCs, even though more study is required. In line with the role of TIPRL in the expression of LC3, CD133, and CD44, we demonstrate the significant upregulation of TIPRL and LC3 in tissues of hepatocyte-derived liver diseases while being downregulated in cirrhosis, the terminal stage of fibrosis. Additionally, we showed the substantial downregulation of CD133 and CD46 in both HCCs and iCCA, and of CD44 in HCCs. Contrarily, we discovered the upregulation of CD133, CD46, and CD44 in liver disease tissues, except for cirrhosis. Therefore, our data suggests that the differential expression of all five variables is associated with disease status in adult hepatocytes-derived liver diseases during disease and cancer progression.

Recently, our team has reported that TIPRL works as a drug-resistant modulator in HCCs via suppressing the MKK7/JNK and the subsequent caspase pathways [3] and accelerates the stemness of liver cancer cells by upregulating the expression of LC3 and CD133 [8]. This role of TIPRL in the stemness of liver cancer cells was further supported by the observation that only TIPRL knockdown reduced the expression of LC3 and CD133 rescued by the overexpression of TIPRL [8]. In keeping with these reports, we demonstrate that TIPRL knockdown decreased LC3, CD133, and CD44 expression and reduced the stemness and self-renewal abilities of huh7, HCCs, and SNU-1097, iCCA, cells. Furthermore, we determine that the TIPRL level was a critical play in the prognosis of liver disease/cancer patients. Therefore, this data indicates that TIPRL is a significant player; moreover, the panels are involved in the liver cancer/HCCs aggravation, even though further study is required, together suggesting the potential usage of the four variables singularly or in conjunction as an early liver biomarker.

Early detection of liver cancer is critical for increasing patient survival [26]. However, a commonly used histological methodology, such as hematoxylin-eosin staining, cannot clearly distinguish between early liver cancer and progenitor lesions of cancer, such as regenerated or dysplastic nodules [27,28]. Furthermore, even though much progress has been achieved, a current screening methodology, such as computed tomography, and a biomarker, such as alpha-fetoprotein (AFP), exhibit low sensitivity and specificity for detecting early liver cancers that can hardly be acceptable in a clinical field [28,29]. Therefore, discovering novel biomarkers for liver cancer that are not influenced by other clinical parameters by determining their roles and prognostic and diagnostic efficacies is urgently required.

Previously, we suggested TIPRL, LC3, and CD133 as suitable biomarker panel(s) for detecting early liver cancers. Here, we show that the variables TIPRL, LC3, CD133, and CD44 each, excluding CD46, are associated with liver disease patients’ survival. Moreover, TIPRL has a critical effect on the patient’s survival of liver diseases, including cancer, determined by our sets and public databases and KM plot analysis. Intriguingly, these relationships were supported by a crucial prognostic potential of the four variables in liver disease patients, determined by ROC analysis. In line with the variables’ acute prognostic effects, we determined a significant association between the levels of the variables, without CD46, in human liver cancer tissues. These significant associations are in agreement with the prominent effect of TIPRL with LC3 and CD133 on diagnostic patients with grade 1 iCCA and with the superior efficiency of four variables, excluding CD46, on prognosticating with grade 1 HCCs and iCCA.

Here, our data implies that the differential expression of all five variables has a crucial effect on HCCs and iCCA, during liver cancer progression. Furthermore, TIPRL could best explain liver cancer patients’ survivability as a sole variable among the rest and contributes to liver cancer cell survival by stemness and self-renewal induction. This independence further supports the significant association between the levels in the variables, excluding CD46, and prominent diagnostic patients with grade 1 iCCA and prognosticating with grade 1 HCCs and iCCA. This study was performed based on the retrospective cohort study with a low number of samples. Therefore, to be adapted for use in the clinical field, our results must be further validated by prospective studies with a high number of samples in clinical trials.

## 4. Materials and Methods

### 4.1. Patients’ Tissues and Information

For a retrospective study based on prospectively collected data, we grouped patients’ samples (OD-CT-DgLiv01-012 and LV1221; US Biomax: Rockville) into the training and the validation set, respectively (Appendix A). We obtained the patients’ information from US Biomax. This study was approved by the Korea Research Institute Bioscience and Biotechnology (KRIBB), and the need for informed consent was waived. Ethical Approval: All human tissues were treated according to the ethical guidelines of the institutional and/or national research committee and the 1964 Helsinki Declaration and its later amendments or comparable ethical stands. We obtained approval for the study protocol from the Institutional Review Board of KRIBB (P01-202104-31-006).

### 4.2. Immunohistochemistry and Histopathology

Patient tissues were analyzed for levels of all five variables, TIPRL, LC3, CD133, CD44, and CD46, as described previously [8]. Briefly slides were deparaffinized and antigen-retrieved using xylene and a series of ethanol (Merck, Darmstadt, Germany, 1.00983.1011) and a pre-heated sodium citrate buffer (0.01 M, pH 6.0; Merck, Darmstadt, Germany, S4641), respectively. The slides were then reacted with antibodies against TIPRL (1:100; Bethyl laboratories, Montgomery, TX, USA, A300-663A), LC3 (1:100; Merck, Darmstadt, Germany, L7543), CD133 (1:100; NOVUS, Centennial, CO, USA, NBP2-37741), CD44 (1:100; Invitrogen, Waltham, MA, USA, MA5-13890), and CD46 (1:200; Santa Cruz Biotechnology, Dallas, TX, USA, sc-52647) overnight. After that, each slide was reacted with Alexa Fluor 533 goat anti-rabbit or Alexa Fluor 488 anti-mouse (Thermo Fisher Scientific, Waltham, MA, A21071, T458, respectively). Quantification of each expression (Appendix A) was determined using the Zeiss program (ZEN 2.3 lite) followed by confocal observation (Zeiss LSM 800). For confirmation of the specificity of the antibodies that used in the study, the stomach and lung tissues were stained with the antibodies against the variables, and then observed under confocal microscope (Appendix A). Our data demonstrates consistency with the reports in “The human protein atlas (https://v15.proteinatlas.org/cancer
www.oncomine.org accessed on 1 June 2021)” regarding the negative responsiveness of TIPRL, CD46, CD133, and LC3 in the stomach, and the positive reactivity of TIPRL, CD46, CD133, and LC3 in lung tissues. We also observed the positive sensitivity of CD44 in both stomach and lung tissues. Because of the difficulty of collecting human tissues for negative staining, we did our utmost to demonstrate the staining specificities using the different tissues that we could obtain.

### 4.3. Statistical Analysis

Continuous variables were represented as the median and standard error of the mean. To determine significance, we used *p*-values with a % difference. The Kaplan–Meier method was used to estimate survival curves, and a difference between the low and the high level for each candidate was determined by the Log-rank test with 95% CI and *p*-values. GraphPad Prism 7 (GraphPad Software, version 5.0, GraphPad Holdings, LLC, San Diego, CA, USA) was used to plot the ROC and determine the AUC with 95% CI, *p*-values, sensitivity, and specificity. The cut-off values for each variable were calculated using the Youden index. A uni- and multivariate Cox proportional hazard model by R version 3.1.0 was used to determine the impact of continuous variables, TIPRL, LC3, CD133, CD44, and CD46 with the categorical variables as covariates on liver disease and/or liver cancer patients’ survival time. The proportionality test determined whether the variables violated the proportional risk assumption. The variable having *p*-values < 0.05 on univariate analysis were selected for multivariate analysis. We used Spearman’s rank correlation coefficient to determine a correlation between continuous variables.

### 4.4. Cell Culture and Small-Interfering RNAs (siRNAs) Transfection

Human HCC cell line, huh7 (ThermoFisher Scientific), ICC cell line, SNU1097 (KCLB) and Chang liver (ATCC) cells were maintained in DMEM or RPMI medium supplemented with 10% fetal bovine serum (FBS; Corning, 35-015-CV) and authenticated by the Korean Collection for Type Cultures. Mycoplasma contamination was tested regularly.

For TIPRL knockdown, four different small-interfering RNAs based on TIPRL sequence (TIPRL, TIP41, NM_152902.3) were constructed, and all exhibited over 90% TIPRL knockdown efficiencies. Among them, the #3 construct was mainly used for the current study; 5′-CCUAAUGAAAUAUCCCAGUAUUU-3′ (sense), 5′-AUACUGGGAUAUUUCAU UAGGUU-3′ (anti-sense). siCont (universal control; STPharm, South Korea) was 5′-AUGAACGUGAAUUGCUCAATT-3′ (sense) and 5′-UUGAGCAAUUCACGUUCAUTT-3′ (anti-sense). For the others’ knockdown, siRNA sequences used were provided in Appendix A. siRNAs were transfected at a final concentration of 100 nmole/L for 72 h with Lipofectamine RNAmaxi (Thermo Fisher Scientific, Waltham, MA, USA 13778150) according to the manufacturer’s protocols.

### 4.5. Cell Proliferation and Survival Assays (MTT Assay)

To determine the effect of siRNAs knockdown on proliferation and survival of Huh7, SNU1097, and Chang cells, the MTT (M2128, Merck, Darmstadt, Germany) assay was performed. Briefly, 24 h after siRNAs transfection, the transfected cells were reseeded onto either attached plates (353072, Corning, Glendale, AZ, USA) or suspension plates (7007, Corning, Glendale, AZ, USA). After 48 h, the media was replaced with 2 mg/mL MTT solution. The plates were then incubated in the dark for additional four hours. MTT solution was removed, followed by addition of DMSO (1380, Duksan, South Korea) to dissolve the dark blue formazan precipitates. Optical density was measured using a microplate reader (Multiscan Go, Thermo Scientific, Waltham, MA, USA) at 570 nm.

### 4.6. Reverse-Transcriptase Quantitative Polymerase-Chain Reaction

RNA extraction and the following reverse-transcription (RT) with 1.0 μg RNA and Oligo (dT)_12-18_ primers were performed using a PureHelix^TM^ Total RNA purification kit (Nanohelix, Daejeon, South Korea) and HelixCript^TM^ First cDNA Synthesis Kit (Nanohelix, Daejeon, South Korea), respectively. A polymerase-chain reaction (PCR) was conducted with the specific primers for the target genes as follows (Appendix A): the RT and the qPCR were carried out using the GeneAmp PCR system 9700 (Applied Biosystems) and the CFX96TM Real-Time System (Bio-Rad, Daejeon, South Korea) in a 10 μL reaction mixture containing 1 μL of diluted DNA template, 2 pmole of each primer and either 5 μL of HelixAmp^TM^ Ready-2x-MultiPlex or RealHelixTM Premier qPCR Kit (Nanohelix, Daejeon, South Korea), respectively. For the control, glyceraldehyde 3-phosphate dehydrogenase (GAPDH) was used. qPCR amplification was carried out three times independently. Each PCR product for the target genes was confirmed as a single band of the expected size on 1.5% agarose gel.

### 4.7. Tumorspheres Formation

Cells (3 × 10^5^ cells/well for huh7 cells; 5 × 10^5^ cells/well for SNU1097 cells) were seeded in ultra-low affinity plates and maintained in serum-free DMEM/RPMI1640 containing 20 ng/mL of human epidermal growth factor and 10 ng/mL of human basic fibroblast growth factor (PeproTech EC, London, UK, AF-100-B and 100-18B, respectively). Individual spheres > 50 μm from each replicate wells (*n* ≥ 9 wells) were counted under an inverted microscope at 40× magnification using the ImageJ program (The National Institutes of Health). The percentage of cells capable of forming spheres, termed “tumorsphere formation efficiency (TSFE),” was determined as follows: [(the number of tumorspheres formed/the initial number of cells seeded) × 100].

## 5. Conclusions

In summary, the variables TIPRL, LC3, CD133, and CD44 reflect the overall survival of liver cancer patients. TIPRL has the most critical effect on liver cancer patients’ survivability as a sole covariate and a prominent efficiency on diagnostic grade 1 iCC and prognosticating grade 1 HCCs and iCCA. This efficiency further supports that only the depletion of TIPRL significantly reduced cell viability and the self-renewal ability of HCCs and iCCA cell lines via the decrease of stemness-related genes.

## Figures and Tables

**Figure 1 cancers-13-02925-f001:**
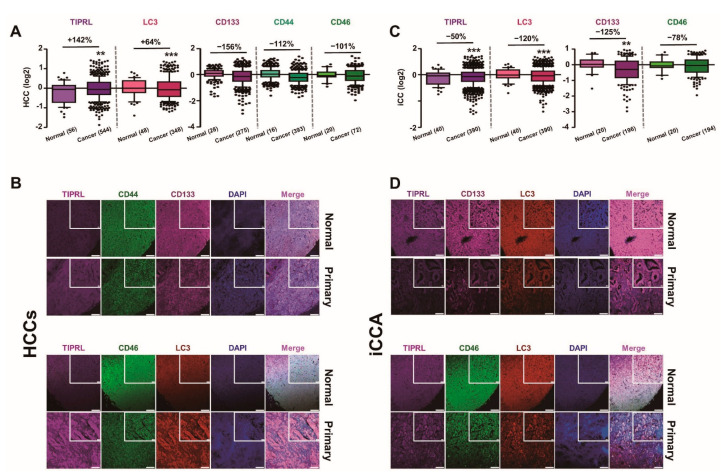
Differential expression of all five variables, depending on liver cancer cell types. Human liver cancer tissues were stained with the indicated antibodies followed by confocal observation. (**A**,**C**) The levels of all five variables were obtained using the ZEN program, and then global normalization was carried out (Appendix A). *p*-values (** *p* < 0.01; *** *p* < 0.001) were determined by a paired *t*-test, and % differences are shown. (*n*) is the number of samples. (**B**,**D**) The images represent normal and HCCs (**B**)/iCCA (**D**), respectively. DAPI was used for nucleus staining, and scar bars, 20 μm (inserted), and 100 μm.

**Figure 2 cancers-13-02925-f002:**
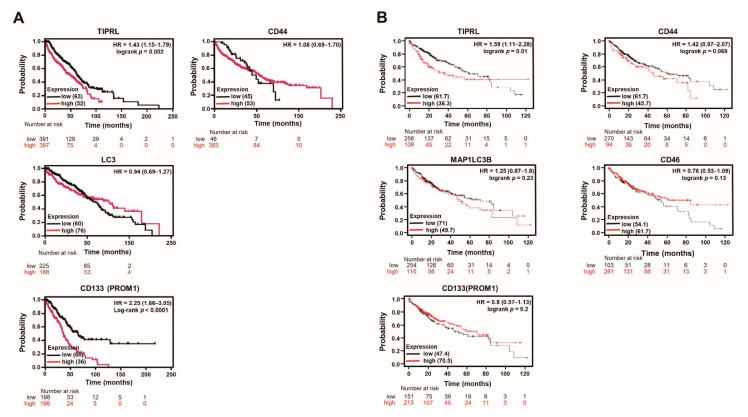
TIPRL predicts the poor prognosis of liver disease patients. (**A**) The survival time of liver disease patients in the training set was calculated using the Kaplan–Meier estimator. HR, (95%CI), Log-rank and *p*-values are noted. (**B**) A public database (www.kmplot.com accessed on 12 January 2021) shows the survivability of liver cancer patients in each variable.

**Figure 3 cancers-13-02925-f003:**
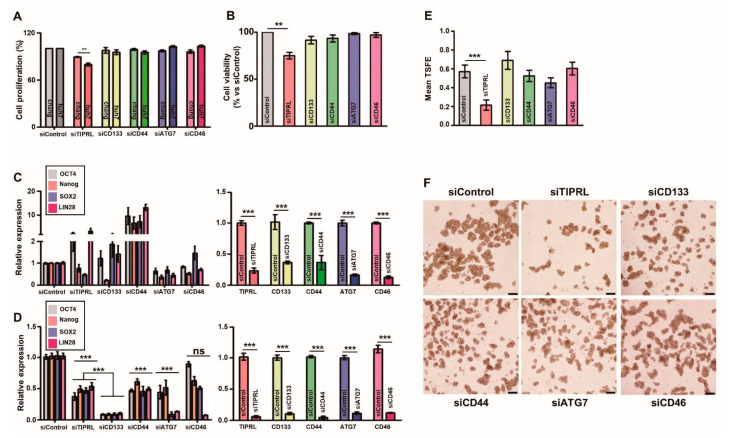
The prominent role of TIPRL in huh7, HCC, cell viability and stemness. Chang (**A**,**C**) and huh7 (**A**–**F**) cells were cultured, and then cells were seeded in a 96-well plate (**A**) or Anoikis plates (**B**,**D**–**F**) followed by transfected with the indicated siRNAs (100 nM). For cell proliferation (**A**) and viability (**B**) assays, 48 h after siRNA transfection, an MTT assay was performed. For quantification analysis of expression in stemness-related genes, we performed RT-qPCR using primers (**C**,**D**; Appendix A). (**E**,**F**) We counted the numbers of spheroids after 72 h siRNAs transfection. TSFE, tumorsphere formation efficiency: [(the number of tumorspheres formed/the initial number of cells seeded) × 100]. All experiments were independently repeated three times. ** *p* < 0.01, *** *p* < 0.001 by unpaired *t*-test. ns, not significant.

**Figure 4 cancers-13-02925-f004:**
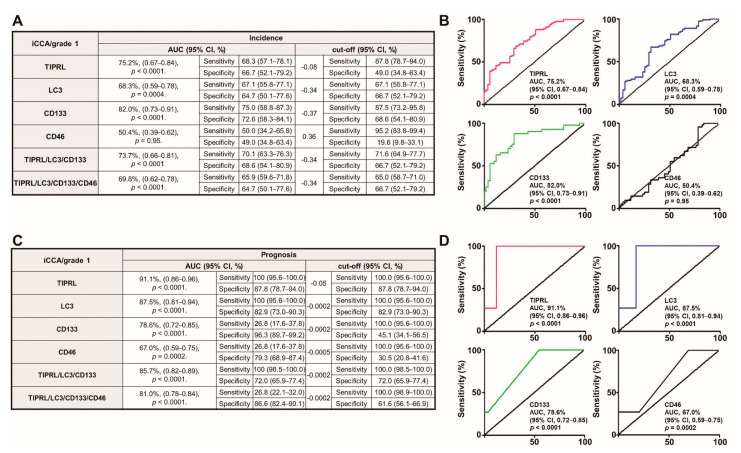
The significant diagnostic and prognostic efficacies of the variables, TIPRL, LC3, and CD133, excluding CD46, as a single or in conjunction for grade 1 iCCA. (**A**,**B**) The diagnostic and (**C**,**D**) prognostic potentials of the variables were determined using ROC analysis. AUC, the area under the curve, and CI, confidence interval. The number of cancer grade 1 used = 82.

**Table 1 cancers-13-02925-t001:** Univariate analysis of overall survival of liver disease patients according to possible prognostic variables Analysis of (A) the training and (B) of the validation set. Abbreviations: HR = hazard ratio; CI = confidence interval; ref = reference.

A. Training			B. Validation		
**Variables**	Overall Survival		**Variables**	Overall Survival	
	Univariate			Univariate	
	HR (95 % CI)	*p*-values		HR (95 % CI)	*p*-values
**Age**			**Age**		
≤50	ref		≤50	ref	
>50	0.57 (0.40–0.80)	0.001	>50	0.88 (0.64–1.22)	0.44
**Sex**			**Sex**		
Female	ref		Female	ref	
Male	0.58 (0.40–0.83)	0.003	Male	3.0 × 10^−^^7^ (5.47 × 10^−^^14^-1.65)	0.96
**Subtypes**			**Subtypes**		
Hepatocellular carcinoma	0.79 (0.47–1.33)	0.37	Hepatocellular carcinoma	4.72 (1.72–13.01)	0.003
Chronic hepatitis	0.57 (0.33–0.97)	0.04	Intrahepatic cholangiocarcinoma	3.91 (1.44–10.63)	0.008
Cirrhosis	1.72 (0.96–3.08)	0.07	Mixed carcinoma	1.93 (1.47–16.53)	0.01
Hepatic steatosis	2.38 (1.18–4.79)	0.02	Carcinoid	55.72 (12.89–240.92)	7.36 × 10^−8^
Liver cell/tissue degeneration, Inflammation of portal area	NA		Adenosquamous carcinoma	4.30 (1.07–17.39)	0.04
Normal liver tissue	ref		Cancer adjacent liver tissue	ref	
**TNM**			**TNM**		
T1N0M0	2.13 (1.49–3.05)	3.83 × 10^−5^	T2N0M0	2.84 (1.06–7.65)	0.04
T2N0M0	1.24 (0.97–1.58)	0.08	T2N1M0	2.32 (1.09–4.92)	0.03
T2N1M0	NA		T3N0M0	2.28 (0.68–7.67)	0.18
T2N0M1	8.5 × 10^−7^ (0.0-Inf)	0.99	T3N1M0	1.95 (0.95–4.02)	0.07
T3N0M0	0.38 (0.24–0.61)	5.43 × 10^−5^	T3N0M1	5.12 (1.89–13.88)	0.001
T4N0M0	NA		T3N1M1	3.52 (0.44–28.29)	0.24
**Grade**			T4N1M0	1.36 (0.41–4.56)	0.62
Grade1	0.32 (0.19–0.53)	1.58 × 10^−5^	T4N1M1	3.62 (1.24–10.59)	0.02
Grade 1-2	0.28 (0.16–0.47)	1.47 × 10^−6^	**Grade**		
Grade2	0.19 (0.13–0.29)	1.89 × 10^−15^	Grade1	0.89 (0.52–1.52)	0.67
Grade 2-3	0.11 (0.06–0.19)	3.06 × 10^−16^	Grade2	0.92 (0.62–1.37)	0.67
Grade3	NA		Grade3	1.84 (1.21–2.80)	0.005
**Stage**			**Stage**		
stage1	0.37 (0.18–0.78)	0.009	stage1	NA	
stage2	0.20 (0.09–0.41)	1.38 × 10^−5^	stage2	2.76 (1.03–7.42)	0.04
stage3	0.02 (0.009–0.06)	<2 × 10^−16^	stage3	2.17 (0.65–7.30)	0.21
stage4	7.844 × 10^−8^ (0.0-Inf)	0.99	stage4	2.11 (1.03–4.30)	0.04
**Markers**			**Markers**		
TIPRL	1.36 (1.04–1.78)	0.02	TIPRL	1.53 (1.26–1.87)	2.46 × 10^−5^
LC3	0.60 (0.42–0.86)	0.005	LC3	1.04 (0.83–1.31)	0.72
CD133	1.50 (0.96–2.34)	0.07	CD133	1.52 (1.01–2.10)	0.01
CD44	1.21 (0.91–1.61)	0.19	CD46	0.92 (0.63–1.34)	0.67

**Table 2 cancers-13-02925-t002:** Multivariate analysis of overall survival of liver disease patients according to potential prognostic variables multivariate analysis of overall survival of liver disease patients according to the significant prognostic variables on univariate analysis in the training (A) and the validation (B) set. Abbreviations: HR = hazard ratio; CI = confidence interval; ref = reference.

A. Training			B. Validation		
**Variables**	**Overall Survival**		**Variables**	**Overall Survival**	
	Univariate			Univariate	
	HR (95 % CI)	*p*-values		HR (95 % CI)	*p*-values
**Subtypes**			**Subtypes**		
Hepatocellular carcinoma	0.21 (0.10–0.41)	8.31 × 10^−6^	Hepatocellular carcinoma	6.89 (2.18–21.79)	0.001
Chronic hepatitis	0.43 (0.25–0.74)	0.002	Intrahepatic cholangiocarcinoma	5.25 (1.74–15.85)	0.003
Cirrhosis	1.91 (1.05–3.47)	0.04	Mixed carcinoma	5.13 (1.53–17.23)	0.008
Hepatic steatosis	2.32 (1.14–4.71)	0.02	Carcinoid	59.86 (13.81–259.43)	4.53 × 10^−8^
Liver cell/tissue degeneration, Inflammation of portal area	NA	NA	Adenosquamous carcinoma	4.46 (1.10–18.08)	0.04
Normal liver tissue	ref		Cancer adjacent liver tissue	ref	
**TNM**			**TNM**		
T1N0M0	5.68 (3.10–10.40)	1.88 × 10^−8^	T2N0M0	2.25 (0.83–6.11)	0.11
T2N0M0	3.84 (2.29–6.45)	3.64 × 10^−7^	T2N1M0	2.64 (1.22–5.71)	0.01
T2N1M0	NA		T3N0M0	3.04 (0.86–10.76)	0.08
T2N0M1	2.2 × 10^−6^ (0.0-Inf)	0.99	T3N1M0	2.10 (1.01–4.37)	0.05
T3N0M0	NA		T3N0M1	7.71 (2.79–21.29)	8.13 × 10^−5^
T4N0M0	NA		T3N1M1	1.70 (0.21–13.70)	0.62
			T4N1M0	1.49 (0.44–5.04)	0.52
			T4N1M1	4.15 (1.37–12.55)	0.01
			**Grade**		
			Grade1	0.63 (0.33–1.21)	0.17
			Grade2	0.57 (0.32–1.01)	0.06
			Grade3	1.22 (0.69–2.16)	0.50
			**Stage**		
			stage1	NA	
			stage2	2.13 (0.78–5.78)	0.14
			stage3	2.67 (0.76–9.39)	0.13
			stage4	2.27 (1.10–4.69)	0.03
**Markers**			**Markers**		
TIPRL	2.06 (1.52–2.79)	2.77 × 10^−6^	TIPRL	14.65 (8.34–25.74)	<2 × 10^−16^
LC3	0.29 (0.17–0.51)	1.60 × 10^−5^	LC3	0.03 (0.02–0.07)	<2 × 10^−16^
CD133	0.63 (0.37–1.06)	0.08	CD133	3.56 (2.17–5.84)	4.70 × 10^−7^
CD44	2.23 (1.41–3.51)	0.0006	CD46	2.13 (1.47–3.08)	6.16 × 10^−5^

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
