# Peer review of "The Human TOR Signaling Regulator Is the Key Indicator of Liver Cancer Patients’ Overall Survival: TIPRL/LC3/CD133/CD44 as Potential Biomarkers for Early Liver Cancers"

_cancers, 2021, doi:10.3390/cancers13122925_

Round 1

Reviewer 1 Report

The efforts taken by the authors to revise the manuscript is appreciated. 

I recommend acceptance of the manuscript for publication. Please check for minor errors and language corrections. 

Reviewer 2 Report

The authors politely responded to each revise in new revision and cover letter.

This manuscript is a resubmission of an earlier submission. The following is a list of the peer review reports and author responses from that submission.

Round 1

Reviewer 1 Report

The manuscript of Jun et al., addresses the potential of selected biomarkers as a prognostic markers for liver cancer. The manuscript is well written, extensively validated and the efforts taken by the authors to present this research study is appreciated. 

The study excels in utilizing public domain available data sets, use of patient samples and in the confirmation using in vitro approaches. 

Figure 1 : It is very difficult to understand the staining specificity of the selected markers from the images provided as there is no differences in the staining patterns except for the level of intensities. Did the authors see any structural abnormalities correlative with the cancer grade. This can be addressed by visualization of the confocal images at higher magnification. It is also recommended to test the staining specificities using a negative control. 

Figure 4 - While the main focus has been on grade -1 cancer, have the authors checked the relationship with other grades. 

While the data presented is conclusive of the identified markers in grade-1 liver cancer, there is little information on the association with other grades. Does the importance of the identified markers holds true when the analysis is done with respect to grade 2 or 3. Try analyzing and presenting Figure 4 as presented in table 1. This will further establish the importance of the identified markers. 

Author Response

Thank you for allowing us to amend our manuscript entitled, "The human TOR signaling regulator is the key indicator of liver cancer patients' overall survival: TIPRL/LC3/CD133/CD44 as potential biomarkers for early liver cancers." 

Briefly, in our revised manuscript, considering the comments, we uploaded the original images (100 µm) with their higher magnification pictures (20 µm) to show their correlation between their abnormalities and cancer grade/ morphological type. Consistent with cancer grade 1, we determined the statistically acceptable prognostic ability of the four variables in cancer grades 2/3 iCCs. A commonly used methodology, such as hematoxylin-eosin staining, cannot clearly distinguish between early liver cancer and progenitor lesions of cancer; thus, biomarkers for detecting early liver cancers are performed as an additional test in the clinical field. However, a currently used biomarker, such as AFP, has reported its low performance in the clinical field. Therefore, we believe that the reviewer's comments were successfully addressed. Consequently, we ask that you consider the enclosed manuscript for publication.

Reviewer 2 Report

Comments to the authors

The manuscript entitled “The human TOR signaling regulator is the key indicator of liver cancer patients' overall survival: TIPRL/LC3/CD133/CD44 as potential biomarkers for early liver cancers” by Soo Young Jun, et al. reported that TIPRL/LC3/CD133/CD44 could, either individually or in conjunction, serve as potential biomarkers for early liver cancer. Although the study is of interest and important, there are some flaws to be revealed.

Major comments

  1. Authors revealed that TIPRL predicted the poor prognosis of liver disease patients in Figure 2. Did the liver disease patients include HCCs or iCCs? If so, is there any influences of treatments such as surgical procedures or chemotherapy on the expression of markers in cancerous patients?

  1. The paragraph (Line 270-276) does not fit into the Results section. The authors should describe at Discussion section.

  1. The authors showed the significant prognostic efficacy of the variables, TIPRL, LC3, and CD133, excluding CD46 as a single or in conjunction for grade 1 iCCs in Figure 4C and D. It seems that there are some flaws in the analysis of prognostic efficacy of these variables in Figure 4D, how many patients are there grade 1 iCCs?

  1. As authors mentioned, the variables, TIPRL, LC3, CD133, and CD44 reflected the overall survival of liver cancer patients. The authors should more discuss the impact of TIPRL or TIPRL/LC3/CD133/CD44 and how they clinically use these potential biomarkers for early liver cancers.

Minor comments

  1. The authors should unify the colors of legends between A and B or C and D in Figure 1. It is very confusing.

  1. The authors should unify “ICCs” and “iCCs” in Line 140.

  1. The authors should correct “Figure 7C-D” to “Figure S7C-D”.

  1. The authors should correct the Bold format (Line 325-326 and 343).

  1. The authors should delete one phrase “DSS Disease-specific survival”.

Author Response

Thank you for allowing us to amend our manuscript entitled, "The human TOR signaling regulator is the key indicator of liver cancer patients' overall survival: TIPRL/LC3/CD133/CD44 as potential biomarkers for early liver cancers." 

Briefly, given our and others' previous reports, we believe that chemotherapy could increase the expression of the variables, singularly or in conjunction. A commonly used methodology, such as hematoxylin-eosin staining, cannot clearly distinguish between early liver cancer and progenitor lesions of cancer; thus, biomarkers for detecting early liver cancers are performed as an additional test in the clinical field. However, a currently used biomarker, such as AFP, has reported its low performance in the clinical field. Therefore, we believe that the reviewer's comments were successfully addressed; TIPRL is a significant player. Moreover, the panels are involved in liver cancer/HCCs aggravation, suggesting the potential usage of the four variables, singularly or in conjunction, as an early liver biomarker. Consequently, we ask that you consider the enclosed manuscript for publication.

Please note that we denoted revisions in red to assist the reviewer.  

Reviewer 3 Report

The study by Jun and co-workers explored the clinical significance of TIPRL in predicting cancer progression and patient prognosis in HCC and ICC.  The authors also revealed in vitro that TIPRL worked as an upstream for LC3 and CD133, and thereby contributed to liver cancer cell proliferation, viability, and stemness.  Moreover, the authors demonstrated that co-evaluation of TIPRL/LC3/CD133/CD44 might be a useful biomarker for the diagnosis and predicting prognosis in early liver cancer.  The author concluded that it is useful to explore the precise mechanisms of liver cancer progression induced by TIPRL/LC3/CD133 signaling.  The concept of this study for focusing on TIPRL/LC3/CD133 signaling is interesting; however, a lot of serious issues are raised concerning to this study.

1) The authors showed the univariate and multivariate analyses of overall survival of liver disease in relation to clinicopathological findings and several prognostic markers using the training set and the validation set; however, the results were obviously different between the training set and the validation set.  In addition, the ROC data predicting prognosis is also very different between the training set and the validation set.  Therefore, it seemed that the reliability of these results was low.  The authors mentioned that these differences might be based on the difference in the rate of ICC between the training set and the validation set.  If the authors believe this hypothesis, the authors should evaluate the data of HCC patients and ICC patients, separately. 

2) The authors demonstrated that TIPRL expression was upregulated in HCC, and downregulated in ICC.  However, the authors also found that TIPRL knockdown directly inhibited cell proliferation and indirectly suppressed stem cell-like property and self-renewal capacity by inhibiting TIRPL/LC3/CD133 pathway in both HCC cells and ICC cells in vitro.  Generally, malignant potential such as stem cell-like property and self-renewal capacity is higher in ICC compared to HCC.  The authors’ results showed that TIPRL enhanced cancer progression, but decreased in clinical ICC samples.  How the authors explain these discrepancies?  For clinical use, do we need to inhibit the function of TIPRL in ICC, even though TIPRL expression is low in ICC?

3) In the present study, the authors showed increased expression of TIPRL and decreased expression of CD133; however, previous reports by the authors revealed that TIPRL enhanced tumor malignancy via upregulation of LC3/CD133 in HCC.  The authors mentioned that the percentage of G1 phase is different (10% vs 0%).  However, CD133 was decreased in more than 10% of patients.  The authors should show another reason why enhanced TIPRL expression did not upregulate CD133 expression in clinical HCC samples in this study.  The authors need to show the relationship between TIPRL expression and LC3/CD133 expression in clinical HCC and ICC samples separately using immunofluorescence data.  Moreover, it is interesting if the authors evaluate the differences in the clinicopathological data between high TIPRL expression and low TIPRL expression.  The authors are also requested to show the changes in the expression of LC3/CD133 after TIPRL knockdown in HCC cells and ICC cells in vitro.

4) The authors mentioned that combined evaluation of TIRPL/LC3/CD133 expression is useful for the early diagnosis of HCC.  I assume that needle biopsy of the tumor is needed for evaluating the expression of these factors.  If needle biopsy is performed, it seems that H-E staining is easy and useful for the diagnosis of HCC.  The authors need to show the merit of combined evaluation of TIRPL/LC3/CD133 expression, compared to the evaluation with only H-E staining.

5) The authors also demonstrated that combined evaluation of TIRPL/LC3/CD133/CD44 expression reflected patient prognosis in early ICC and HCC.  Generally, the prognosis of early stage liver cancer, especially HCC, is good.  The authors should discuss about the clinical usefulness of predicting the prognosis of patients with early liver cancer using the combined evaluation of TIRPL/LC3/CD133/CD44 expression.

Author Response

Thank you for allowing us to amend our manuscript entitled, "The human TOR signaling regulator is the key indicator of liver cancer patients' overall survival: TIPRL/LC3/CD133/CD44 as potential biomarkers for early liver cancers." 

Briefly, in our revised manuscript, considering the comments, we addressed that a commonly used methodology, such as hematoxylin-eosin staining, cannot clearly distinguish between early liver cancer and progenitor lesions of cancer. Thus, biomarkers for detecting early liver cancers are performed as an additional test in the clinical field. However, a currently used biomarker, such as AFP, has reported its low performance in the clinical field. This study demonstrates a significant cancer grade-dependent association between TIPRL and LC3/CD133/CD44 in HCCs and iCCs. Therefore, we believe that the editor and reviewer's comments were successfully addressed; TIPRL is a significant player. Moreover, the panels are involved in liver cancer/HCCs aggravation, suggesting the potential usage of the four variables, singularly or in conjunction, as an early liver biomarker. Consequently, we ask that you consider the enclosed manuscript for publication.

Please note that we denoted revisions in red to assist the reviewer.  
